# Household costs, catastrophic out-of-pocket payments and impoverishment related to accessing surgical care in rural Ethiopia

Yohannes Hailemichael[1], Tigist Eshetu[2], Sewit Timothewos[2], Andualem Deneke[3], Amezene Tadesse[3], Ahmed Abdella[4], Abebe Bekele[3], Andrew Leather[5], Girmay Medhin[6], Martin Prince[7], Charlotte Hanlon[2,8]*

1 Armauer Hansen Research Institute, Addis Ababa, Ethiopia, 2 Centre for Innovative Drug Development and Therapeutic Trials for Africa (CDT-Africa), College of Health Sciences, Addis Ababa University, Addis Ababa, Ethiopia, 3 Department of Surgery, School of Medicine, College of Health Sciences, Addis Ababa University, Addis Ababa, Ethiopia, 4 Department of Obstetrics and Gynaecology, School of Medicine, College of Health Sciences, Addis Ababa University, Addis Ababa, Ethiopia, 5 King's Centre for Global Health and Health Partnerships, School of Life Course and Population Sciences, King's College London, London, United Kingdom, 6 Aklilu-Lemma Institute of Pathobiology, Addis Ababa University, Addis Ababa, Ethiopia, 7 King's Global Health Institute, King's College London, London, United Kingdom, 8 Division of Psychiatry, Centre for Clinical Brain Sciences, University of Edinburgh, Edinburgh, Scotland, United Kingdom

* chanlon@ed.ac.uk

## Abstract

### Background

The objective of this study was to assess the household costs, catastrophic out-of-pocket (OOP) health expenditure, impoverishment and coping mechanisms used to pay for surgical care in a predominantly rural area of Ethiopia.

### Methods

We conducted a community-based, cross-sectional household survey of 182 people who had undergone a surgical procedure. Participants were interviewed in their homes at six weeks post-operation. Using a contextually adapted version of the Study of global AGEing and adult health (SAGE) questionnaire, we estimated direct and indirect costs of surgical care from a household perspective. Catastrophic out-of-pocket (OOP) health expenditure was estimated using thresholds of 10% and 25% of annual household consumption expenditure. Impacts of surgical care payments on poverty levels was estimated by comparing pre- and post-operative OOP payments. Analysis of variance, t-test and a logit model were used to assess factors associated with catastrophic OOP health expenditure.

### Results

Most surgical patients were women (87.9%), with 65.0% receiving obstetric surgical care. Direct costs dominated expenditure: direct average medical costs Birr 1649.5

**Data availability statement:** The data underlying the results presented in the study are available from OSF: https://osf.io/439ar/files/osfstorage/66f9436980cf6639363cbc89.

**Funding:** The research underpinning the analyses in this paper was supported by the National Institute for Health and Care Research (NIHR) Global Health Research Unit on Health System Strengthening in Sub-Saharan Africa (ASSET), King's College London (GHRU 16/136/54 to MP, CH, AA, AB and AL) using UK aid from the UK Government.

**Competing interests:** The authors have declared that no competing interests exist.

**Abbreviations:** ASSET: Health System Strengthening in sub-Saharan Africa; OOP: Out-of-pocket; SAGE: Study on global Ageing and adult health; SDGs: Sustainable development goals; UHC: Universal health coverage; WHO: World health Organization.

(44.6%), direct average non-medical costs Birr 1226.5 (33.2%), indirect average costs Birr 821.9 (22.2%). Catastrophic OOP surgical care expenditure was experienced by 69.2% households at the 10% threshold and 45.6% at the 25% threshold. The increase in average normalized poverty gap due to OOP surgical care expenditure was higher in non-obstetric (14.1%) compared to obstetric (5.8%) procedures, and for non-emergency (13.3%) compared to emergency care (6.3%). To pay for surgical care, 38.0% of households had sold assets and 5.0% had borrowed money.

## Conclusions

Due to surgical care, households faced severe financial burdens leading to impoverishment. To mitigate the resulting financial constraints, households implemented hardship coping strategies. Provision of free obstetric surgical care reduced, but did not eliminate, these burdens. There is a pressing need to tailor financial risk protection mechanisms to achieve universal coverage for surgical care.

## Background

In low-income and lower-middle-income countries, over nine out of ten people cannot access safe, timely and affordable surgical and anaesthesia care [1]. Key barriers include the direct and indirect costs related to surgical services [2,3]. Surgical conditions additionally can have a severe impact on the person's day-to-day activities and result in a loss of income to the household, either in the form of wage losses from the patient or caregivers, or due to losses in agricultural output or other earnings. Shrime et al. (2016) reported that half of the world population, or 3.7 billion people (posterior credible interval: 3.2–4.2 billion), are at risk of catastrophic expenditure if they were to need surgery because they do not have financial risk protection [4]. Each year, an estimated 81 million people worldwide are driven to financial catastrophe due to the cost of surgery and its associated indirect costs, with the biggest burden borne by people in low- and middle-income countries (LMICs) [5]. In patients who underwent surgery at a government hospital in Uganda, 31% faced catastrophic expenditure [6]. In Malawi, 94% of district and 87% of central hospital patients experienced catastrophic expenditure for surgical care [7].

In Ethiopia pre-payment and other financial risk protection mechanisms are limited. In recognition of this, in recent years the country has introduced health care financing reforms to increase affordable access to health services and achieve universal health coverage (UHC) [8]. Efforts to enhance financial risk protection for people accessing essential health services in Ethiopia include provision of high-impact interventions free of charge through an exemption program; subsidization of more than 80% of the cost of care in public health facilities; implementation of community-based health insurance (CBHI) schemes; and full subsidization of the very poor through fee waivers [9]. However, the coverage of CBHI is low (28%) [10], while the coverage of social health insurance for the formal sector is less than 3% [11] and fee waivers are based on a quota system, covering only 10% of the poorest population

[12]. Therefore, for the majority, care provision is based on "fees for service". Thus, out-of-pocket health expenditure still dominates all sources of expenditures for health care and constituted 35% of total health expenditure in 2018 [13]. As a result, OOP healthcare expenditure has impoverishing effects on households of those who experience ill-health, especially among the poorest [14,15]. In addition, due to the impact of OOP payments, a large treatment gap is likely to exist. In a recent study from Ethiopia, Kiros et al. (2020) estimated that, on average, 0.9% (range 0.1–5.0%) of the population becomes impoverished every year because of the cost of accessing health care services [16]. Thus, where healthcare payments are made mostly through OOP payments, it becomes a trade-off for patients and families to get care and suffer catastrophic expenditure or accept the consequences of not accessing care. As is the case in Ethiopia, many households may risk not seeking care [17].

Several studies in Ethiopia have investigated and compared the burden of OOP health expenditure and impoverishment for specific health conditions [18–20] and between different regional states [15,16]. However, we were not able to find any studies on the magnitude of catastrophic health care expenditure due to surgical care. It is important for policy makers to understand the financial impact of surgical care on the economic welfare of households to guide the national health policy to achieve equitable, accessible and affordable surgical services. In light of this, the objective of this study was to quantify direct and indirect costs, investigate the magnitude of catastrophic out-of-pocket payments and impoverishment related to accessing surgical care and to identify the financial coping strategies implemented by affected households.

## Materials and methods

We conducted a community-based, cross-sectional household survey of people who had undergone a surgical procedure and were interviewed in their homes at six weeks after the operation. This study was conducted as part of the pre-implementation phase of the ASSET programme (Health System Strengthening in sub-Saharan Africa) [21]. The goal of ASSET was to identify system bottlenecks to accessing quality surgical, maternal and primary healthcare in Ethiopia, Sierra Leone, South Africa and Zimbabwe [22]. In this paper, we present findings from Ethiopia.

### Study setting

The study was conducted in Meskan, Misrak Meskan, Sodo and South Sodo districts and Butajira town of the Gurage zone located in the Southern Nations, Nationalities and People's Region of Ethiopia. Health services are provided through governmental and non-governmental health facilities. In the five districts there are 94 health posts and 16 primary health care units (health centers) for preventive and curative health services. There is one general and one primary hospital where surgical care is available. In the general hospital, there is one surgeon, one obstetrician/gynaecologist and two Integrated Emergency Surgical Officers (IESOs) who provide both emergency and elective surgical care. In the primary hospital, there are two IESOs providing only emergency surgical care. The common surgical procedures performed in the hospitals are Caesarean section, abdominal hysterectomy, management of pelvic organ prolapse, tubal ligation, repair of hernias, management of acute appendicitis, haemorrhoidectomy, fracture and dislocation management and tooth extractions. Most surgical care services are based on a "fee for service" model, but obstetric care is an exempted service.

### Participants

Participants were all surgical patients who had undergone a surgical procedure between 3rd June and 27th September 2019 at the general or primary hospital and were available for a community-based survey 6–8 weeks following the operation. The aim was to investigate the costs incurred when accessing surgical care and to estimate the proportion of people facing catastrophic OOP payments and impoverishment (pushing the household below the poverty threshold) [21,23].

### Recruitment procedure

Participants were approached by research staff while in hospital once their condition had stabilized. They were invited to participate in the study and provide informed consent for research staff to visit their home for an interview on expenditures they encountered related to surgical care.

### Sample size

We estimated that we would recruit 150–200 participants over the four months study period. This was a pragmatic decision based on what was feasible and comparable with previous studies from Malawi [7] and Ghana [24].

### Outcome variables

The primary outcome was cost of surgical care, with secondary outcomes of catastrophic OOP healthcare expenditure, the poverty impact of accessing surgical care and coping strategies implemented to manage financial constraints.

### Measurements

A contextually adapted and abbreviated version of the World Health Organization SAGE (Study of global AGEing and adult health) survey instrument was used. SAGE was used previously in a study on health and ageing in six LMICs [25] and in Ethiopia [14]. The SAGE instrument is used to collect information on a variety of individual and household socio-economic attributes such as consumption expenditure, income, assets and household demographics (detailed below). Cost of illness was derived from consumption expenditure data and income loss of patients and caregivers associated with seeking surgical care and providing care for the person who had undergone a surgical procedure.

Type of surgery was categorised as emergency versus elective with the assumption that this would be relevant to the cost of care. Moreover, we classified the type of surgical care as obstetric vs. non- obstetric procedures because only obstetric care was exempted from full costs. These classifications were based on information obtained from hospital records. Sociodemographic data were measured in terms of age, sex, residential area (urban/rural), marital status, religion and education. Information on the availability of financial risk protection mechanisms for the patient was obtained from the hospital record and classified as community-based health insurance (CBHI), a 'free certificate' (fee waiver) or not covered by any financial risk protection mechanism.

### Consumption expenditures

Consumption expenditure variables were collected at the household level. Food expenditure was estimated by summing consumption of the household's own food products and expenditure on food items over the seven days preceding the survey. Non-food expenditure was calculated by summing expenditure on rent, electricity, transport, clothes and health care for the 30 days preceding the survey, and expenditure on education, health aids, hospitalization and long-term health care for the 12 months preceding the survey. To obtain comparable time periods, all consumption expenditures data were annualised and then converted to a 30-day time period. Monthly household consumption expenditure was calculated as the sum of food, non-food and health expenditure.

### Out-of-pocket health expenditure

Out-of-pocket health expenditures were those made by households at the point of receiving health services and payments directly related to seeking care (transportation, food, accommodation and caregiving) as reported by respondents. We further classified OOP payments into 'catastrophic' and 'non-catastrophic' categories. Catastrophic OOP health expenditure was defined in relation to a household's consumption expenditures. OOP health care payments were classified as financially catastrophic when they exceeded 10% or 25% of total household consumption [26,27].

For cost of surgical care, data were collected on expenditures incurred as follows:

(i) direct medical costs, including registration, consultation, medicines, diagnostics (investigations and procedures), medical supplies and hospital charges; (ii) direct non-medical costs, like transportation to seek care for the affected person and accompanying household members, food, accommodation, hiring of someone as a care provider/attendant; and (iii) indirect costs, which was income that was foregone due to the patient's inability to work or other income foregone by other household members because they accompanied the patient to the facility or spent time caring for the patient at home [28]. In estimating productivity losses, we followed recommended practice to use the actual income losses rather than the potential losses. The rationale is that, in agricultural societies or for people engaged in informal labour, there are seasons in which work intensity is high and others in which work intensity is low. In addition, household use coping strategies that may mitigate these potential losses. Thus, days of ill-health do not necessarily translate neatly into days of lost work [29,30]. Therefore, participants were asked how much money they and/or their caregiver had lost during the illness because of not participating in productive activities or income generation. The values reported by the respondent were considered to be indirect costs to the patient and caregiver. All costs were expressed in Ethiopian Birr and exchange rate to USD and PPP (the purchasing power parity conversion rate to the international dollar) was provided as a footnote to the tables) [31,32].

### Incidence and intensity of catastrophic out-of-pocket health expenditure

The incidence of catastrophic OOP health expenditure is measured by catastrophic head count, which is the percentage of households incurring OOP health expenditure in excess of a specified threshold in one year [33]. In this study, two thresholds were used: greater than or equal to 10% and 25% of total consumption expenditure. Both thresholds are indicators linked to Sustainable Development Goals for monitoring Universal Health Care [34].

The intensity of catastrophic OOP health expenditure captures how much a household's OOP health expenditure exceeds the catastrophic threshold and is estimated through the catastrophic overshoot and the mean positive overshoot (MPO) [33]. For calculation of catastrophic OOP healthcare expenditure, direct medical and non-medical costs were used.

### Poverty impact of out-of-pocket health expenditure

The poverty impact of OOP health expenditure was measured in terms of:

(i)   poverty head count, which is the percentage of households living below the poverty line;

(ii)  poverty gap, or the average amount by which resources fall short of the poverty line, which measures poverty depth or intensity of poverty (the amount by which poor households fall short of the poverty line).

(iii) normalized poverty gap, obtained by dividing the poverty gap by the poverty line for OOP health expenditure pre- and post-surgical procedure.

The poverty line was set based on median consumption expenditure, a relative poverty line of half and two-thirds of median consumption per capita [35,36].

### Statistical analysis

OOP health expenditure and poverty impact of surgical care expenditures were analyzed using one-way analysis of variance, Student's t-test and the Kruskal Wallis test depending on the required statistical assumptions and number of categories to be compared. Categorical data were cross-tabulated and the statistical significance of observed differences in proportions was investigated using Pearson's Chi-square ($\chi^2$) or alternatively Fisher's exact test, as appropriate. Further analysis was conducted to investigate the effect of OOP health expenditure on household food and non-food consumption expenditure using a linear probability model. See Supplementary file 4 S4 Table for equations for the statistical analyses.

## Inclusivity in global research

Additional information regarding the ethical, cultural, and scientific considerations specific to inclusivity in global research is included in the Supporting Information S1 File.

## Ethics approval and consent to participation

This study was conducted according to the principles expressed in the Declaration of Helsinki. Ethical approval for the study was obtained from the Institutional Review Board of the College of Health Sciences, Addis Ababa University (026/18/psy; 16th May 2018), and the Research Ethics Committee of King's College London (RESCM-17/18–6144; 18th August 2018). All participants gave written, informed consent. For people who were non-literate, a witness confirmed that the information sheet had been read out correctly and the participant provided a fingerprint.

## Results

Out of 224 people who underwent surgical procedures during the study period, 182 (81.3%) participated in the survey in the community after 6–8 weeks. See Fig 1 for a flow chart of participants.

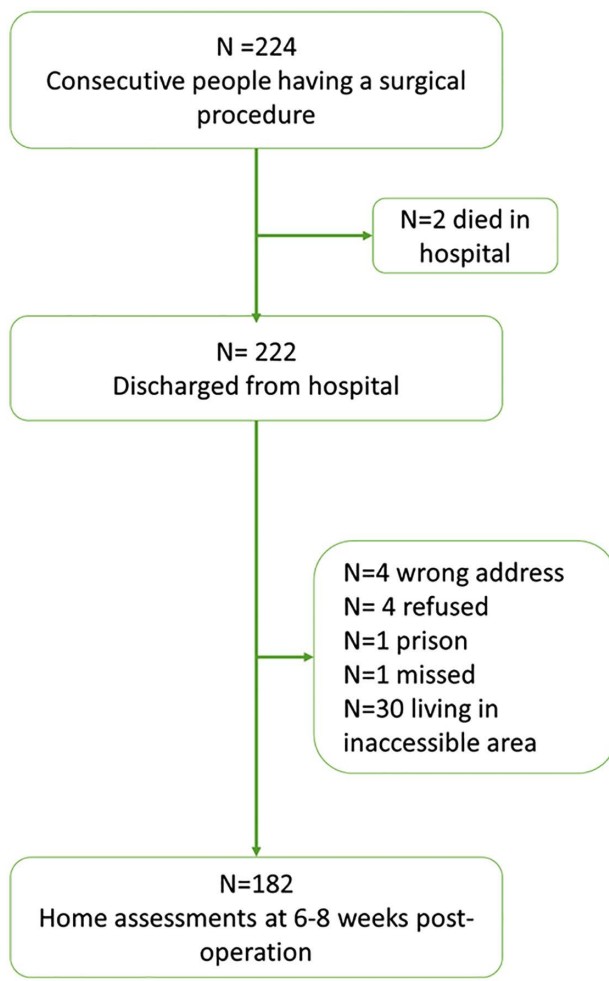

**Fig 1. Flow chart of participants.**

## Background characteristics

See Table 1 for demographic characteristics of respondents. The mean (standard deviation; SD) age of respondents was 33.1 (SD 13.5) years, with 84.1% within the 19–40 years age group. Most (about 88%) of household heads were men, 56.0% (n = 102) were Muslim, 81 (44.5%) had attended primary level education and 55 (30.2%) had no formal education. The majority (87.9%) of respondents were married. In terms of financial protection, 14 (8.4%) of the study participants were members of CBHI and 25 (13.8%) had a poverty certificate (fee waiver) to access health services without payment. The frequencies of differing types of surgical operation are presented in Supplementary File 5 S5 Table: 66% were obstetric surgical procedures.

## Consumption expenditures

Supplementary table 1 S1 Table shows the monthly household consumption expenditure by consumption categories. On average, household food consumption was Birr 5970.3 (SD 4400.3), representing 43.7% of total consumption expenditure, with Birr 2423.3 (SD 7981.7) spent on household utilities, Birr 1694.9 (SD 6830.9) on big purchases and Birr 2876.1 (SD 3848.9) on healthcare.

Table 2 provides the detailed household consumption expenditure by household characteristics and surgery type (emergency vs. elective; obstetric vs. non-obstetric). The share of OOP payments for health care as a percentage of total consumption expenditure was 16.4% for obstetric surgical procedures compared to 35.6% for non-obstetric procedures (p < 0.001). Health care costs for rural residents were about 13% (p < 0.001) higher than health care costs for urban residents. The cost of treatment for women was about 19.7% (p < 0.001) less in terms of OOP health care expenditure compared to the cost of treatment for men.

Households that faced catastrophic OOPHE had significantly lower expenditure on food (p = 0.002), housing utilities (p = 0.008) and big item expenditures (p = 0.012) compared to non-catastrophic households.

## Direct and indirect costs of surgical care

Table 3 shows the mean expenditure on surgical care. Direct medical costs contributed the largest proportion (44.6%) of the economic cost of surgical care, comprising an average of Birr 1649.5 per respondent. Of the direct medical costs, 36.2% was spent on medication and 29.3% on other medical items.

Following direct medical costs, the cost of direct non-medical and indirect costs accounted for an average of Birr 1226.5 and Birr 821.9 per respondent, respectively. The major contributor to direct non-medical expenses was expenditure on food items (mean Birr 596.8). Income loss to the patient and family member accounted for 22.2% of all total costs of surgical care. Table 4 shows the distribution of health care cost categories across households.

There was a similar pattern of distribution with the direct medical, direct non-medical and indirect costs, with males, rural residents, and those undergoing non-obstetric surgery spending more compared to their counterparts. Length of hospitalization had an effect on treatment cost. For every additional day of hospitalization patients incurred Birr 376.0 (on average) for direct medical costs.

## Catastrophic out-of-pocket health care expenditure

Table 5 shows the distribution of catastrophic OOP healthcare expenditure across consumption expenditure quintiles. The proportion of households whose healthcare expenditure was above the two set thresholds for catastrophic spending (i.e., 10% and 25%) was high. For the 10% threshold, catastrophic OOP healthcare expenditure was significantly higher for male patients compared with female patients (95.4% vs. 64.3%; p < 0.001). Similarly, catastrophic OOP health expenditure was higher in those with non-emergency compared to emergency conditions (p = 0.016) and in those undergoing non-obstetric surgical care compared to obstetric care (p < 0.001). Significantly more rural households experienced catastrophic OOP health expenditure compared to urban residents (83.9% vs. 54.1%; p < 0.001). Catastrophic OOP health expenditure was higher in participants who were hospitalized for 6 or more days (93.1%) compared to those who hospitalized for 1–3 days (57.1%; p = 0.005).

**Table 1. Participant characteristics.**

| Characteristics | Number of study participants | Percentage |
|---|---|---|
| Age group (years) | | |
| 19-40 | 153 | 84.1 |
| 41-50 | 8 | 4.4 |
| 51-60 | 9 | 4.9 |
| 61 or above | 12 | 6.6 |
| Gender | | |
| Male | 22 | 12.1 |
| Female | 160 | 87.9 |
| Residence | | |
| Urban | 96 | 54.2 |
| Rural | 81 | 45.8 |
| Education attainment | | |
| No formal education | 55 | 30.2 |
| Primary education | 81 | 44.5 |
| Secondary education | 20 | 11.0 |
| Above secondary | 26 | 16.3 |
| Marital status | | |
| Never married | 11 | 6.0 |
| Married | 160 | 87.9 |
| Separated/ Divorced/ Widowed | 11 | 6.0 |
| Religion | | |
| Muslim | 102 | 56.0 |
| Orthodox Christian | 68 | 37.4 |
| Protestant Christian | 12 | 6.6 |
| Member of Community-Based Health Insurance | | |
| Yes | 14 | 8.4 |
| No | 153 | 91.6 |
| Free health care certificate (fee waiver) | | |
| Yes | 25 | 13.8 |
| No | 156 | 86.2 |
| Either CBHI or fee waiver | | |
| Yes | 39 | 21.4 |
| No | 153 | 78.6 |
| Surgery by condition | | |
| Emergency | 122 | 67.1 |
| Non-emergency | 60 | 32.9 |
| Surgery by type | | |
| Obstetric | 120 | 65.9 |
| Non-obstetric | 62 | 34.1 |
| Length of hospital stay (days) | | |
| 1-3 | 84 | 46.9 |
| 4-5 | 72 | 40.2 |
| 6 or more | 23 | 12.9 |

**Table 2. Mean consumption expenditures (in Birr) by participant characteristics, catastrophic health expenditure and type of surgical care (n = 182).**

| Participant characteristics | Food Mean (%) | Housing utilities Mean (%) | Big expenditures Mean (%) | Healthcare Mean (%) | All expenditures |
|---|---|---|---|---|---|
| Residence | | | | | |
| Urban | 6804.5(46.7) | 3590.3(24.5) | 1700.5(11.6) | 2540.1 (17.3) | 14635.4 |
| Rural | 4905.8(46.2) | 1016.1(9.6) | 1439.4(13.6) | 3241.4 (30.6)*** | 10602.7 |
| Gender of patient | | | | | |
| Male | 6699.4(44.5) | 1210.5(8.0) | 1229.2(8.1) | 5904.9(39.2) | 15044.0 |
| Female | 5795.4(46.8)** | 2514.4(20.3)** | 1589.4(12.8) | 2459.7(19.9)*** | 12358.9 |
| Type of surgery | | | | | |
| Obstetric | 5923.9(45.7) | 2979.6(23.0) | 1927.7(14.9) | 2124.3(16.4) | 12955.5 |
| Non-obstetric | 5867.5(48.2) | 1151.4(9.5) | 806.9(6.7) | 4331.4(35.6)*** | 12157.2 |
| | | | | | |
| Emergency | 6234.6(49.8) | 2073.5(16.6) | 1575.6(12.6)* | 2615.3(17.3) | 12499.0 |
| Non-emergency | 5234.0(40.0)** | 2932.6(22.4) | 1485.5(11.3) | 3406.5(26.0)** | 13058.6 |
| Out-of-pocket expenditure | | | | | |
| Catastrophic | 5263.8(45.7)*** | 1481.6(12.8)*** | 805.5(6.9)*** | 3958.3(34.3)*** | 11509.2 |
| Non catastrophic | 7275.0(47.1) | 4458.3(28.9) | 3128.8(20.2) | 562.6(3.6) | 15424.7 |

*1 USD = 29.07 Birr (2019, purchasing power parity, 1 international dollars=10.74 Birr)*

*ANOVA and t-test for means and Pearson's χ2 for categorical variables. P<0.05*,P<0.01**;P<0.001***.*

*% = percent of expenditure as a share of the four categories*

**Table 3. Cost of surgical care (in Birr) by cost categories (N = 182).**

| Cost categories | Mean (Standard Deviation) | Median (Inter-Quartile Range) | Percentage share |
|---|---|---|---|
| A. Direct medical | 1649.5 (2600.6) | 500 (1980) | 44.6 |
| Registration | 7.3 (8.8) | 10 (10) | 0.4 |
| Medications | 596.8 (1137.3) | 200 (800) | 36.2 |
| Medical supplies | 264.0 (1012.2) | 0 (50) | 16.0 |
| Hospital bed | 225.4 (743.8) | 25 (80) | 13.7 |
| Investigations | 73.3 (201.3) | 0 (50) | 4.4 |
| Other medical items | 482.6 (1212.1) | 15 (450) | 29.2 |
| B. Direct non-medical | 1226.5 (1956.1) | 540 (970) | 33.2 |
| Transportation | 227.3 (611.7) | 75 (135) | 18.5 |
| Food items | 596.8 (937.6) | 310 (550) | 48.7 |
| Caregiver/attendants | 402.5 (1097.4) | 5 (30) | 32.8 |
| C. Indirect cost (lost income) | 821.9 (1698.6) | 400 (1000) | 22.2 |

*1 USD = 29.07 Birr (2019, purchasing power parity, 1 international dollars=10.74 Birr)*

## Impoverishing effect of out-of-pocket payments

As shown in Supplementary table 2 S2 Table, using half and two-thirds of household per capita median consumption expenditures as the poverty line, out-of-pocket surgical care expenditures led to a 10.4% and 19.2% rise in poverty head count ratios, respectively. The normalized poverty gap increased by 6.0% (a 90.9% increment) for a poverty line of half median consumption expenditure, and by 8.6% (an 82.7% increment) for the poverty line of two-thirds median

**Table 4. Mean cost of surgical care (in Birr) by cost category, surgical type and participant characteristics (N=182).**

| Characteristics | Number of study participants | Direct medical cost Mean (%) | Direct non-medical cost Mean (%) | Indirect cost Mean (%) | Total cost |
|---|---|---|---|---|---|
| Gender | | | | | |
| Male | 22 | 3682.2(54.4) | 2222.7(32.9) | 856.8 (12.7) | 6761.7 |
| Female | 160 | 1370.1(41.8) | 1089.6 (33.2) | 817.2(25.0) | 3276.9 |
| P value of t-test | | <0.001 | 0.017 | 0.918 | |
| Residence | | | | | |
| Urban | 96 | 1516.6(46.2) | 1023.5 (31.2) | 740.5(22.6) | 3280.6 |
| Rural | 81 | 1754.1(41.8) | 1487.3 (35.4) | 956.9 (22.8) | 4198.3 |
| P value of t-test | | 0.269 | 0.060 | 0.202 | |
| Type of surgery | | | | | |
| Obstetric | 120 | 1129.7(38.8) | 994.6(34.2) | 786.6 (27.0) | 2910.9 |
| Non-obstetric | 62 | 2655.8(50.9) | 1675.5 (32.0) | 890.3(17.1) | 5221.6 |
| P-value t-test | | <0.001 | 0.025 | 0.697 | |
| | | | | | |
| Emergency | 122 | 1387.2(39.5) | 1228.2(35.0) | 891.9(25.4) | 3507.2 |
| Non-Emergency | 60 | 2183.1(53.4) | 1223.4(30.0) | 679.8(16.6) | 4086.3 |
| P value of t-test | | 0.026 | 0.506 | 0.785 | |
| Out-of-pocket health expenditure | | | | | |
| Catastrophic | 124 | 2351.1 (47.3) | 1607.2 (32.4) | 1008.9 (20.3) | 4967.2 |
| Non-catastrophic | 58 | 149.8(15.2) | 412.8(32.0) | 422.2 (32.8) | 984.8 |
| P value of t-test | | <0.001 | <0.001 | 0.029 | |
| Time in hospital | | | | | |
| 1-3 days | 84 | 905.4(32.8) | 1093.5(39.6) | 761.0(27.6) | 2759.9 |
| 4-5 days | 72 | 1819.2(48.1) | 1144.1(30.3) | 818.8(21.6) | 3782.1 |
| 6 or more days | 23 | 3908.3(55.8) | 2022.8(28.9) | 1075.4(15.3) | 7006.5 |
| P value of ANOVA | | <0.001 | 0.557 | 0.389 | |

*1 USD = 29.07 Birr (2019, purchasing power parity, 1 international dollars=10.74 Birr)*

*% = percent of expenditure as a share of the three categories*

consumption. Similarly, using half of median consumption, the poverty gap increased by Birr 289.9, representing a 92.0% relative increment. A similar pattern was observed when the two-thirds of median total consumption threshold was used as the poverty line.

A stratified analysis of poverty impact of OOP health expenditure for surgical care is presented in Table 6. The analysis shows that using a poverty line of half of median consumption expenditure, 3.9% of obstetric, 17.6% of non-obstetric, 6.6% of emergency and 18.3% of non-emergency patients were pushed below the poverty line after paying for surgical care. The poverty gap also increased by Birr 2042.1 for obstetric surgical intervention, Birr 2119.9 for non-obstetric surgical intervention, Birr 2134 for emergency cases and Birr 2022.8 for non-emergency cases.
Out-of-pocket payments for surgical care raised the average poverty gap (normalized gap) for households of obstetric, non-obstetric, emergency and non-emergency cases by 5.8%, 14.1%, 6.3% and 13.3%, respectively.

At the poverty line of two-thirds of consumption expenditure, 14.1% of obstetric surgical procedures, 29.0% of non-obstetric surgical procedures, 18.0% of emergency and 21.7% of non-emergency patients fell into poverty due to out-of-pocket payments for healthcare. We also found significant differences in the poverty gap after paying for health care. The poverty gap increased by Birr 2184.4 for obstetric surgical procedures, Birr 2510.2 for non-obstetric surgical procedures, Birr 2193.3 for emergencies and Birr 2413.5 for non-emergencies.

**Table 5. Household experiencing catastrophic OOPHE by socio-economic status and surgical type (N = 182).**

| Characteristics | Catastrophic OOP healthcare expenditure measures | | | | | |
| --- | --- | --- | --- | --- | --- | --- |
| | >=10% threshold (95% CI) | | | >=25% (95% CI) | | |
| | Headcount | Overshoot | Mean Poverty Overshoot | Headcount | Overshoot | Mean Poverty Overshoot |
| Gender | | | | | | |
| Male | 95.4***(72.6-99.4) | 31.3***(21.0-41.6) | 32.8(22.5-43.1) | 68.1**(45.9-84.3) | 19.4**(10.7-28.1) | 28.5(18.9-38.0) |
| Female | 64.3(56.5-71.4) | 14.8(11.7-17.9) | 23.0(18.9-27.0) | 37.5(30.2-45.3) | 7.5(5.1-9.8) | 20.0(15.0-24.9) |
| Residence | | | | | | |
| Urban | 54.1(44.0-63.9) | 10.9(7.6-14.2) | 20.1(15.4-24.9) | 31.2(22.7-41.2) | 5.0(2.8-7.2) | 16.0(10.6-21.4) |
| Rural | 83.9***(74.1-90.5) | 22.6***(17.5-27.6) | 26.9(21.4-32.3) | 50.6**(39.7-61.4) | 12.7**(8.5-16.8) | 25.1(18.8-31.3) |
| Type of procedure | | | | | | |
| Obstetric | 56.6(47.5-65.3) | 11.4(8.2-14.5) | 20.1(15.5-24.8) | 27.5 (20.1-36.2) | 15.7(10.7-20.7) | 19.8(13.5-26.2) |
| Non-obstetric | 90.3***(79.8-95.6) | 27.2***(21.3-33.0) | 30.1(24.1-36.1) | 67.7*** (55.0-70.2) | 15.7(10.7-20.7) | 23.2(17.0-29.4) |
| Emergency | 62.2(53.2-70.5) | 13.5(10.3-16.7) | 21.6(17.5-25.8) | 34.4(26.4-43.3) | 6.6(4.3-8.8) | 19.1(14.4-23.9) |
| Non-emergency | 80.0*(67.7-88.3) | 23.5**(17.0-29.9) | 29.3(22.2-36.5) | 55.0**(42.1-67.1) | 13.7(8.3-19.1) | 24.9(16.9-33.0) |
| CBHI | | | | | | |
| Yes | 67.9(60.0-74.9) | 14.8(3.7-25.9) | 20.8(6.5-35.0) | 35.7(15.0-63.5) | 7.3(0.9-15.6) | 20.5(0.2-43.1) |
| No | 71.4(42.6-89.3) | 17.7(14.2-21.6) | 26.0(21.8-30.2) | 43.7(36.0-51.8) | 9.6(6.9-12.3) | 22.0(17.3-26.7) |
| Free certificate | | | | | | |
| Yes | 68.0(47.2-83.4) | 13.9(4.7-23.2) | 20.5(7.9-33.2) | 32.0(16.5-52.7) | 7.1(0.3-14.7) | 22.4(0.5-45.4) |
| No | 67.9(60.1-74.8) | 17.1(13.8-20.4) | 25.2(21.2-29.2) | 42.3(34.7-50.2) | 9.2(6.6-11.7) | 21.7(17.3-26.2) |
| Any financial protection | | | | | | |
| Yes | 66.2(58.0-73.5) | 15.4(12.1-18.6) | 23.2(19.1-27.4) | 40.0(32.2- 48.2) | 7.8(5.3-10.3) | 19.5(14.5-24.4) |
| No | 75.6(59.1-87.0) | 22.2(14.2-30.2) | 29.4(20.3-38.5) | 45.9(30.5-62.1) | 13.4(7.0-19.7) | 29.2*(19.8-38.5) |
| Length of hospital stay | | | | | | |
| 1-3 days | 57.1(46.2-67.3) | 11.6(7.7-15.5) | 20.4(14.6-26.1) | 30.9(21.9-41.7) | 5.5(2.6-8.4) | 17.8(10.0-25.7) |
| 4-5 days | 72.2(60.6-81.4) | 17.6(12.7-22.5) | 24.4(18.6-30.1) | 44.4(33.3-56.1) | 9.2(5.4-13.0) | 20.8(14.1-27.5) |
| 6 or more days | 91.3*(70.2-97.9) | 33.0***22.6-43.4) | 36.2(25.9-46.5) | 69.5(47.8-85.0) | 20.8***(11.8-29.8) | 29.9*(20.0 −39.8) |

P < 0.05*, P < 0.01**; P < 0.001***. P is from Pearson's χ2 and Kruskal walls test; CI: confidence interval. Financial risk protection included, CBHI (community-based health insurance) and those granted free certificate to access health care without payment

## Coping strategies implemented in response to financial constraints

Supplementary table 3 S3 Table shows the source of payments for surgical care. More than half (53.3%) of those who received surgical care paid for the service from their income, with over one-third (37.9%) of households needing to sell items, 15.3% having care paid for by relatives and 4.9% needing to borrow money.

## Factors associated with catastrophic OOP health expenditure

Table 7 presents factors associated with catastrophic OOP health expenditure in excess of either 10% or 25% of the household's consumption expenditure. Patients who received obstetric surgical care had lower odds of catastrophic OOP health expenditure (adjusted Odds Ratio (AOR) 0.1, 95% CI: 0.03, 0.7; p = 0.047) compared to non-obstetric surgical patients. Rural residents had significantly higher odds of catastrophic OOP health expenditure than urban residents at the 10% threshold (AOR 4.2, 95%CI: 1.8, 9.7; p = 0.002) and at the 25% threshold (AOR 1.8, 95%CI: 1.2, 3.9; p = 0.049. At the 10% and 25% thresholds, the odds of facing catastrophic expenditure for those who had no financial risk protection mechanism were (AOR 3.3, 95%CI: 0.7–15.4) and (AOR 2.6, 95%CI: 0.7–8.1) times higher, respectively, compared to those who had financial risk protection. However, these differences are not statistically significant.

**Table 6. Poverty impact of OOPHE using 1/2 and 2/3 of annual median consumption as a poverty line by surgical type (N = 182).**

| Poverty measures | Surgical type | | | |
|---|---|---|---|---|
| | Obstetric | Non-Obstetric | Emergency | Non-emergency |
| **I. using 1/2 of median consumption** | | | | |
| Headcount | | | | |
| Prepayment head count | 17.5% | 20.9% | 14.7% | 26.7% |
| Post payment head count | 21.4% | 38.7% | 21.3% | 45.0% |
| Absolute point change | 3.9% | 17.6%*** | 6.6% | 18.3%* |
| Relative change | 22.2% | 84.2% | 44.8% | 68.5% |
| Poverty gaps | | | | |
| Prepayment poverty gap (Birr) | 315.6 | 313.9 | 268.2 | 410.2 |
| Post payment poverty gap (Birr) | 493.5 | 820.6 | 454.7 | 910.2 |
| Absolute gap change(Birr) | 177.9 | 506.6* | 186.5 | 500.0* |
| Relative change | 56.3% | 161.3% | 69.5% | 121.8% |
| Normalized gaps | | | | |
| Prepayment normalized gap$^X$ | 6.6% | 6.5% | 5.6% | 8.6% |
| Mean positive pre-payment poverty gap(Birr) | 1803.4 | 1497.2 | 1817.9 | 1538.4 |
| Post-payment normalized gap$^Y$ | 10.3% | 17.2% | 9.5% | 19.0% |
| Mean positive post-payment poverty gap (Birr) | 2042.1 | 2119.9 | 2134.0 | 2022.8 |
| Absolute percentage point change (impact)$^Z_{(=Y-X)}$ | 3.7% | 10.6%* | 3.9% | 10.4%* |
| Relative percentage change (=Z/X*100) | 56.0% | 163.0 | 69.6% | 120.9% |
| **II. Using 2/3 of median consumption** | | | | |
| Headcount | | | | |
| Prepayment head count | 30.0% | 27.4% | 24.6% | 38.3% |
| Post payment head count | 44.1% | 56.4% | 42.6% | 60.0% |
| Absolute point change | 14.1% | 29.0%* | 18.0% | 21.7% |
| Relative change | 47% | 105.8% | 73.1% | 56.6% |
| Poverty gaps | | | | |
| Prepayment poverty gap | 655.3 | 1025.8 | 539.3 | 925.2 |
| Post payment poverty gap(Birr) | 688.2 | 1586.8 | 943.5 | 1772.9 |
| Absolute gap change(Birr) | 132.9 | 561.0 | 404.2 | 847.7 |
| Relative change | 20.2% | 54.6% | 74.9% | 91.6% |
| Normalized gaps | | | | |
| Prepayment normalized gap$^X$ | 10.3% | 10.8% | 8.4% | 14.5% |
| Mean positive pre-payment poverty gap(Birr) | 2184.4 | 2510.2 | 2193.3 | 2413.5 |
| Post-payment normalized gap$^Y$ | 16.1% | 29.4% | 14.8% | 27.9% |
| Mean positive post-payment poverty gap(Birr) | 2322.7 | 2810.9 | 2213.6 | 2954.8 |
| Absolute percentage point change (impact)$^Z_{(=Y-X)}$ | 5.8% | 14.1%* | 6.3% | 13.3%* |
| Relative percentage change (=Z/X*100) | 56.3% | 130.5% | 75.0% | 91.7% |

*1 USD = 29.07 Birr (2019, purchasing power parity, 1 international dollars=10.74 Birr)*

*Kruskal-Wallis comparing pre and post payment by surgical care sub groups*

## Discussion

This study examined the cost of surgical care among service users in primary and general hospitals in rural Ethiopia. Even though our service-using sample was likely to include households who were already of better socioeconomic status than those not accessing care at all, we found that the cost of surgical care was catastrophic for the majority of households.

**Table 7. Factors associated with catastrophic OOPHE.**

| Characteristics | Catastrophic thresholds | | | |
| --- | --- | --- | --- | --- |
| | >=10% | | >=25% | |
| | Crude Odds Ratio (95%,CI) | Adjusted Odds Ratio (95%,CI) | Crude Odds Ratio (95%,CI) | Adjusted Odds Ratio (95%,CI) |
| Gender | | | | |
| Male | Reference | Reference | Reference | Reference |
| Female | 0.08* (0.01- 0.6) | 0.5(0.04-6.0) | 0.2(0.1-0.7) | 0.6(0.3-5.0) |
| Residence | | | | |
| Urban | Reference | Reference | Reference | Reference |
| Rural | 4.4*** (2.1-9.0) | 4.2** (1.8-9.7) | 2.2**(1.2-4.1) | 1.8* (1.2-3.9) |
| Surgical condition | | | | |
| Emergency | Reference | Reference | Reference | Reference |
| Non- emergency | 2.42* (1.16- 5.0) | 1.0(0.3-3.2) | 2.3**(1.2-4.3) | 1.6(0.2-2.0) |
| Surgical type | | | | |
| Obstetric | 0.1* (0.05-0.3) | 0.1* (0.03-0.7) | 0.1***(0.09-0.3) | 0.1** (0.04-0.5) |
| Non-obstetric | Reference | Reference | Reference | Reference |
| Users by payment status | | | | |
| CBHI | | | | |
| Yes | Reference | Reference | Reference | Reference |
| No | 0.8(0.2-2.8) | 3.1(0.5-18.1) | 1.4(0.4-4.3) | 4.1(0.8-19.9) |
| Free certificate | | | | |
| Yes | Reference | Reference | Reference | Reference |
| No | 0.9(0.4-2.4) | 2.8(0.4-24.1) | 1.5(0.6-3.8) | 2.1(0.3-12.4) |
| Length of hospital stay | | | | |
| 1-3 days | 0.1(0.02-0.5) | 0.3(0.05-2.0) | 0.1**(0.03-0.4) | 0.1* (0.07-0.5) |
| 4-5 days | 0.2(0.05-1.1) | 0.5(0.1-3.8) | 0.5*(0.01-0.8) | 0.3* (0.1-0.9) |
| 6 or more days | Reference | Reference | Reference | Reference |
| Financial risk protection | | | | |
| Yes | Reference | Reference | Reference | Reference |
| No | 0.6(0.2-1.4) | 3.3(0.7-15.4) | 0.7(0.3-1.6) | 2.6(0.7-8.1) |

*Financial risk protection included, CBHI (community based health insurance) or granted free certificate to access health care without payment; p<0.05\*, p<0.01\*\*, p<0.001\*\*\**

More than two-thirds of surgical patients faced catastrophic OOP health expenditure at the 10% threshold of total consumption expenditure. Our finding is lower than reports from Malawi [7] and Madagascar [37], while being greater than the 59% for Vietnam [38], 12% for Sierra Leone [39] and 31% reported for Uganda [6]. All surgical care is considered free in Malawi and Uganda, and in Madagascar there is a form of insurance for the poor [37]. In Ethiopia, surgical services are based on payments at the point of care, with the exception of obstetric services which are meant to be provided free of charge through the exemption programme [9,40,41]. Beyond these differences in healthcare financing strategies across countries, it is likely that the proportion of those in need of surgical care who even manage to access hospital care may vary, expected to be lower in this rural, low-income Ethiopian setting. Thus, those accessing surgical care represent households of relatively higher socio-economic status or women accessing obstetric care. A further reason for varying estimates of catastrophic OOP health expenditure is due to methodological differences in estimating catastrophic

expenditure. In some of these studies, wage loss was included in the estimation of catastrophic health expenditure whereas we considered this cost not to be an OOP payment.

Despite obstetric care being considered an essential and fee-exempted health service at government health facilities in Ethiopia, 56.6% and 27.5% of women that received obstetric surgical care encountered catastrophic OOP payments at the 10% and 25% thresholds, respectively. Earlier studies from Ethiopia reported that maternal services are supposed to be provided free of charge in Ethiopia but costs incurred by the households to assess care are substantial [40,41]. Similarly, in our analysis, despite having financial protection mechanisms either in the form of CBHI or a free certificate to access health care, about 68% of surgical patients faced catastrophic OOP payments. The free health initiative in Sierra Leone provided more protection than in Ethiopia [39]. Nonetheless, our finding is similar to the findings from Ghana that reported 58–87% of insured patients face financial catastrophe when accessing surgical care [24]. There have been similar reports from a previous study from Ethiopia on general health care among the members of CBHI [42]. However, it remains unclear why households that are members of health insurance programs are paying OOP expenses for services and facing catastrophic health expenditure. Probably the frequent stock out of medicines and medical supplies in government facilities may force patients to buy these items from private retail pharmacies. This has been also reported in other studies from Ethiopia [40,41]. On the other hand the fee waiver targeting poor individuals proved inadequate, benefiting only 10% of the poor households [43].

Our findings demonstrated that OOP payment was a major cause of poverty in patients receiving surgical care. Thus, using half and two-thirds of median consumption expenditure as poverty lines, paying for surgical care pushed 10.4 and 19.2 percent of households into poverty, respectively. These findings are higher than the nine percent reported for Sierra Leone [44] and the three percent for Uganda [6]. One reason could be in Sierra Leone patients were from high wealth quintiles and less likely to be affected by OOP payments. The other reason for the difference could be due to the differences in the poverty measures used. We used a relative poverty line while the Sierra Leone and the Uganda studies used international poverty lines. Using a relative poverty threshold might better identify individuals with limited resources that are at risk of having adverse socio-emotional outcomes [45]. Our study might have overestimated the poverty impact of OOP payment for surgical care; the reference household expenditure for calculating catastrophic healthcare expenditure was based on expenditure after surgery which may already have reduced as a financial coping strategy.

In a stratified analysis of poverty impact, both poverty headcount and poverty gap were much higher for non-obstetric and non-emergency patients compared to obstetric and emergency patients, resulting in a major impoverishment burden. The findings for non-emergency vs. emergency surgical conditions reflect that most emergency surgical conditions are obstetric and thus exempted from point-of-care costs. The poverty head count impact of OOP payment using half of median-consumption expenditure for those who had obstetric or non-obstetric surgical care were 3.9 and 17.6%, respectively, while for emergency vs. non-emergency surgical care it was 6.6 and 18.3%, respectively. When a poverty line of two-thirds of median consumption expenditure was used, poverty impact of OOP payment showed similar trends with higher numbers of households being pushed below the poverty line.

The poverty gap following OOP payments increased for all surgical types. However, it was three times higher for non-obstetric than obstetric and twice as high for non-emergency compared to emergency surgical care. This finding indicates that all types of surgical care should be considered in priority setting and when designing financial risk protection mechanisms.

Health care expenditure was the second largest consumption expenditure category for the household both in absolute terms and as a proportion of total consumption expenditure. Spending a large share of household budget on health care might impact expenditure on other subsistence needs of the household. This result is similar to findings from Vietnam [46] and China [47].

In our analysis all respondents incurred OOP spending, irrespective of their payment status (i.e., member of CBHI, free certificate and payer). Overall, patients spent a higher share of surgical care cost on direct medical costs (44.6%), while

the indirect cost (lost income) contributed the smallest share (22.2%). The interviews took place quite early in the recovery phase which might under-estimate the share of indirect costs. A study of surgical in-patient care costs in Sierra Leone reported that direct medical costs accounted for 63% of surgical care costs whereas indirect costs were only 21% [44]. Medications costs were the largest expenditure category of direct medical costs which accords with findings from Rwanda [48], Ghana [24] and Malawi [49].

Income losses of patients and their caregivers due to surgical illness were substantial. However, the observed loss of income was smaller than that reported in previous studies [7,44]. This could be due to the high proportion of our sample that was women (88%) and housewives, thus reporting less income in our study or differences in wage rates. Transportation costs for the patient and accompanying family members required to stay with the patient during their hospital stay also contributed to catastrophic OOP payments. In a review of surgical care costs, Okoroh (2021) identified that transportation costs and wage losses were major drivers of catastrophic expenditures for surgery [50].

Inequitable impacts of surgical care costs were evident in our study. The ratio of health expenditure to total consumption expenditure was highest (30.6%) in rural compared to urban (17.3%) households. This indicates that rural households are disproportionately burdened by the cost of surgical care. This is explained by the high transport cost to access care from rural areas and the lower total household consumption of rural residents. In a household survey of catastrophic health expenditure in India [51] and Zambia [52], the highest proportion of catastrophic expenditure was reported among the poorest and rural residents, although all economic status groups experienced high catastrophic costs. Therefore, financial risk protection mechanisms must do more to target rural households.

In our study, non-emergency or non-obstetric surgical patients incurred significantly more costs compared to emergency or obstetric cases. However, the majority of our sample had undergone obstetric procedures and were exempted from costs at the point of care and had in most cases had presented as an emergency. For such women, transport costs were subsidized or free if they were able to access an ambulance. Our finding of lower surgical care costs for obstetric care is similar with the study from Uganda that reported patients who received Caesarean sections spent less [6], which is explained by the shorter duration of obstetric admissions in the hospital. In our analysis, women paid lower surgical care costs compared to their male counterparts. Longer hospital stays were associated with increased direct medical costs. The longer the stay, the higher the costs of medicines, hospital bed payments and the medical supplies used by the patient. In similar studies, prolonged hospitalization was associated with increased costs [44,53].

Direct medical, direct non-medical and indirect costs were significantly higher for households that faced catastrophic OOP payments for surgical care compared to those households that did not face catastrophic payments. Therefore, these costs are likely to compromise the consumption of food and other subsistence needs and impact negatively on the quality of life of these households.

In practice, in Ethiopia, surgical care service charges are similar regardless of economic status. Hence, the poor pay the same amount for surgical care despite having less income. Therefore, even relatively low spending on surgical care by the poor is catastrophic and thus likely to compromise access to services. Moreover, in Ethiopia where one-fifth of the population is living below the poverty line [9], OOP payment is a major obstacle to the use of essential surgical care and exacerbates impoverishment.

In order to pay for surgical care households used different coping strategies. The majority used their income, which may not be an option for poorer households who could thus not access surgical care at all. In addition, use of income for meeting health expenses reduces current consumption of other goods and services. Moreover, the other coping strategies used were borrowing and selling assets (i.e., hardship coping mechanisms), which are deeply regressive and could induce long-term household economic impacts and impoverishment. Previous studies on coping strategies for paying of surgical care reported similar findings [7,44,50].

Although we believe this is the first study in Ethiopia to investigate surgical care costs, catastrophic expenditure, and poverty impact of OOP payments, our study has the following limitations. Only two hospitals were chosen and these may

not be representative of all hospitals in Ethiopia. The sample size was relatively small and this may have limited the statistical power to detect differences. The sample size was not calculated based on a hypothesis but instead based on what was feasible and aligned with previous studies. The possibility of underestimation of non-medical costs due to recall bias cannot be ruled out. In subsistence farming estimating lost income and consumption expenditure is difficult and might lead to under-estimation of these costs for the respondents. We did not adjust consumption expenditure for household composition (i.e., age and size) as data were not collected on these variables. Thus, estimates were calculated per household which may affect interpretation of the difference between urban and rural households. For both obstetric and surgical care, patients might have undergone minor or major surgery which may have different cost outcomes. However, we were not able to stratify these groups to estimate cost. Lastly, the survey was conducted a few weeks after the patient was discharged from hospital and households may have already reduced expenditure on non-essential (and some essential) items to manage the catastrophic costs. This might have over-estimated the degree to which the surgical costs were catastrophic.

## Conclusions

Due to surgical care, households faced severe financial burdens of treatment costs and lost income leading to impoverishment. This was particularly marked for non-obstetric procedures which were not exempt from payment at the point of care. Households were forced to use hardship coping mechanisms to pay treatment costs. Therefore, surgical care should be accessible to all without financial hardship to achieve universal health coverage. In addition, the Ethiopian health insurance service needs to explore strategies to include direct non-medical expenses like transportation costs that are not covered currently.

## Supporting information

**S1 Table. Household consumption expenditure cost categories.**
(DOCX)

**S2 Table. Poverty impact of OOP payments for surgical care.**
(DOCX)

**S3 Table. Sources of payment for surgical care.**
(DOCX)

**S4 Table. Equations for statistical analyses.**
(DOCX)

**S5 Table. Types of surgical procedure.**
(DOCX)

**S1 File. Inclusivity-in-global-research-questionnaire ASSET.**
(DOCX)

## Acknowledgments

The ASSET research team would like to dedicate this paper to the memory of Professor Amezene Tadesse, who sadly passed away in October 2025. Professor Tadesse was a co-investigator on the ASSET project and an active contributor to surgical system strengthening in Ethiopia and this manuscript. May his soul rest in peace.

We would like to thank all participants in the study for their participation. For the purposes of open access, the author has applied a Creative Commons Attribution (CC BY) licence to any Accepted Author Manuscript version arising from this submission.

Charlotte Hanlon (CH) and Martin Prince (MP) are supported by an NIHR global health research group on homelessness and mental health in Africa (HOPE; NIHR134325). The views expressed in this publication are those of the authors and not necessarily those of the NHS, the National Institute for Health and Care Research or the Department of Health and Social Care, England. CH is also funded by the Wellcome Trust through grants 222154/Z20/Z (SCOPE) and 223615/Z/21/Z (PROMISE). The funders had no role in study design, data collection and analysis, decision to publish, or preparation of the manuscript

## Author contributions

**Conceptualization:** Abebe Bekele, Andrew Leather, Martin Prince, Charlotte Hanlon.

**Data curation:** Tigist Eshetu, Sewit Timothewos.

**Formal analysis:** Yohannes Hailemichael, Girmay Medhin.

**Funding acquisition:** Abebe Bekele, Andrew Leather, Martin Prince, Charlotte Hanlon.

**Investigation:** Sewit Timothewos, Andualem Deneke, Amezene Tadesse.

**Methodology:** Yohannes Hailemichael, Ahmed Abdella, Abebe Bekele, Andrew Leather, Girmay Medhin, Martin Prince, Charlotte Hanlon.

**Project administration:** Tigist Eshetu, Andualem Deneke, Ahmed Abdella, Charlotte Hanlon.

**Supervision:** Tigist Eshetu, Sewit Timothewos, Amezene Tadesse.

**Writing – original draft:** Yohannes Hailemichael.

**Writing – review & editing:** Tigist Eshetu, Sewit Timothewos, Andualem Deneke, Amezene Tadesse, Ahmed Abdella, Abebe Bekele, Andrew Leather, Girmay Medhin, Martin Prince, Charlotte Hanlon.

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
