## [Decision Letter · Decision Letter 0]

1 Feb 2024

Dear Dr. Hanlon,

Thank you for submitting your manuscript to PLOS ONE. After careful consideration, we feel that it has merit but does not fully meet PLOS ONE’s publication criteria as it currently stands. Therefore, we invite you to submit a revised version of the manuscript that addresses the points raised during the review process.

We look forward to receiving your revised manuscript.

Kind regards,

Peivand Bastani

Academic Editor

PLOS ONE

Journal Requirements:

3. Please include a complete copy of PLOS’ questionnaire on inclusivity in global research in your revised manuscript. Our policy for research in this area aims to improve transparency in the reporting of research performed outside of researchers’ own country or community. The policy applies to researchers who have travelled to a different country to conduct research, research with Indigenous populations or their lands, and research on cultural artefacts. The questionnaire can also be requested at the journal’s discretion for any other submissions, even if these conditions are not met.  Please find more information on the policy and a link to download a blank copy of the questionnaire here: https://journals.plos.org/plosone/s/best-practices-in-research-reporting. Please upload a completed version of your questionnaire as Supporting Information when you resubmit your manuscript.

5. Thank you for stating the following financial disclosure: "The research underpinning the analyses in this paper was supported by the National Institute for Health and Care Research (NIHR) Global Health Research Unit on Health System Strengthening in Sub-Saharan Africa (ASSET), King’s College London (GHRU 16/136/54) using UK aid from the UK Government. Charlotte Hanlon (CH) receives support through an NIHR RIGHT grant (NIHR200842); CH and MP are supported by an NIHR global health research group on homelessness and mental health in Africa (HOPE; NIHR134325). The views expressed in this publication are those of the authors and not necessarily those of the NHS, the National Institute for Health and Care Research or the Department of Health and Social Care, England. CH is also funded by the Wellcome Trust through grants 222154/Z20/Z (SCOPE) and 223615/Z/21/Z (PROMISE)."

6. In the online submission form, you indicated that the data underlying the results presented in the study are available from the corresponding author on request

7. Your ethics statement should only appear in the Methods section of your manuscript. If your ethics statement is written in any section besides the Methods, please move it to the Methods section and delete it from any other section. Please ensure that your ethics statement is included in your manuscript, as the ethics statement entered into the online submission form will not be published alongside your manuscript.

Reviewers' comments:

Reviewer's Responses to Questions

**Comments to the Author**

1. Is the manuscript technically sound, and do the data support the conclusions?

Reviewer #1: Partly

2. Has the statistical analysis been performed appropriately and rigorously?

Reviewer #1: No

3. Have the authors made all data underlying the findings in their manuscript fully available?

Reviewer #1: Yes

4. Is the manuscript presented in an intelligible fashion and written in standard English?

Reviewer #1: No

Reviewer #1: Overall comments: Many thanks for giving me the opportunity to read and review this paper. The paper deals with a common issue of healthcare financing for surgical care and estimate the impact on households in terms of various financial indicators. One of the major limitations of this paper that it does not clearly define that what it includes in ‘surgical care’ term. Cost of surgical care has been evaluated without any classification and only provided a board category of those cases considered. I doubt the impact of different types of surgical care will be different on households based on their extent such as for minor surgery or major surgery.

Specific comments:

1. p7,l132: it is not clear why patients were included in the costing if fee is exempted for obstetric care? Will it not affect the total cost per patient?

2. p8, L145: No clear strategy of sample size estimation or justification. Mere mention of reference does not sufficiently justify the sample size used for estimation of cost in this study.

3. p10, l195: Need a clear explanation of the methodology used for estimation of the indirect cost during illness of a patient. How was the total days loss estimated considering the situation that days of ill health do not necessarily translate into days of lost work? How was it decided how many days of a ill patient will be considered as days lost?

4. Table 2. It is not clear what the household characteristics mean in terms of expenditure? How was expenditure differentiated by male or female?

5. All tables should report n for each specific item. For example, from table 3 it is not clear which expenditure item was applicable for how many patients. What happened if any item of expenditure was zero to an individual? How does this representative to actual scenario.

6. As OOP expenditure for health tends to be highly skewed, Table 3 should also include the median expenditure.

7. The author should include equation for each of the estimation procedure to make it clear that how those estimates were derived.

8. The author should also include clear explanation which of statistical test conducted for which of the variables. E.g., Table 6, not clear about the statistical inference.

9. P26, L426: The author mentioned that “Inequitable impacts of surgical care costs were evident in our study.” How the authors become sure that this was evident not due to the number of different types of surgical cases between the two regions?

10. The inclusion of subsidized cases and obstetric cases (the costs of which are also exempted) might have influenced the overall costs estimates per patient. In such cases, how useful it is to conclude the findings on the financial indicators without having a proper comparison of between subsidized cases and non-subsidized cases.

Minor comments.

1. the use of abbreviations is inconsistent throughout the manuscript.

**Do you want your identity to be public for this peer review?** For information about this choice, including consent withdrawal, please see our Privacy Policy

Reviewer #1: No

---

## [Author Response · Author response to Decision Letter 1]

16 Oct 2024

Dr Emily Chenette

Editor-in-chief, PLoS One Journal

Dear Dr Chenette

2nd October 2024

Ref: PONE-D-23-35112.

Costs, catastrophic out-of-pocket payments and impoverishment related to accessing surgical care among households in rural Ethiopia

We would like to express our sincere gratitude for the editor and review comments on our paper and for making allowance for our delayed response. We have responded to each query below. We would be happy to provide further clarifications as needed.

Yours sincerely

Professor Charlotte Hanlon

Editorial points

Response 1: We have now done this.

2. Consider depositing your raw data in a repository to ensure your work is read, appreciated and cited by the largest possible audience.

Response 2: We have now made the dataset underpinning the analyses available via OSF: https://osf.io/439ar/files/osfstorage/66f9436980cf6639363cbc89

3. Please include a complete copy of PLOS’s questionnaire on inclusivity in global research in your revised manuscript.

Response 3: We have added this as a supporting information file and included a section in the methods, as recommended.

Response 4: In the Funding Information we acknowledge the funder of this specific research project. In the financial disclosure section, we additionally acknowledge salary support. Please advise how you would like this to be presented.

Response 5: We confirm that the statement is correct: “The funders had no role in study design, data collection and analysis, decision to publish, or preparation of the manuscript.”

6. In the online submission form, you indicated that the data underlying the results presented in the study are available from the corresponding author on request

Response 6: We have now made the dataset underpinning the analyses available via OSF: https://osf.io/439ar/files/osfstorage/66f9436980cf6639363cbc89

7. Your ethics statement should only appear in the Methods section of your manuscript. If your ethics statement is written in any section besides the Methods, please move it to the Methods section and delete it from any other section.

Response 7: We have amended as requested.

Reviewer points

8. Need for English editing

Response 8: The paper has now been edited by a native English speaker.

9. Overall comments: Many thanks for giving me the opportunity to read and review this paper. The paper deals with a common issue of healthcare financing for surgical care and estimate the impact on households in terms of various financial indicators. One of the major limitations of this paper that it does not clearly define that what it includes in ‘surgical care’ term. Cost of surgical care has been evaluated without any classification and only provided a broad category of those cases considered. I doubt the impact of different types of surgical care will be different on households based on their extent such as for minor surgery or major surgery.

Response 9: Thank you very much for the hugely valuable and constructive comments, which have helped us to improve the manuscript.

On pp 8 and 9 line 160-162 the rationale for categorizing surgical care was described. We acknowledge that both obstetric care and surgical care included major surgery and minor surgery. It is evident that costs may vary between major and minor surgery, Not stratifying cost by these groups is a limitation of the study.

We have therefore added the following text to the limitation “For both obstetric and surgical care, patients might have undergone minor or major surgery which may have had different cost outcomes. However, we were not able to stratify these groups to estimate cost”. PP27 line 480-82

10. p7,l132: it is not clear why patients were included in the costing if fee is exempted for obstetric care? Will it not affect the total cost per patient?

Response 10: The purpose of the economic impact of surgical care is to understand both financial (monetary) and non-financial (the opportunity cost) impact on individuals and their household.

While obstetric care is theoretically provided free of charge in Ethiopia, there are substantial costs associated with accessing the service that limit access, including transportation expenses, income losses, and the opportunity costs of participating in other productive activities. Beyond this, there may be hidden direct costs if medications or materials are not available in the public sector and need to be purchased by the patient/their family. It is with this intention why we included the obstetric patients in the study.

11. p8, L145: No clear strategy of sample size estimation or justification. Mere mention of reference does not sufficiently justify the sample size used for estimation of cost in this study

Response 11: The sample size was not calculated based on a hypothesis but instead based on what was feasible and aligned with previous studies. This is a limitation and we included it in the discussion section. PP 28, line:482-483

12. p10, l195: Need a clear explanation of the methodology used for estimation of the indirect cost during illness of a patient. How was the total days loss estimated considering the situation that days of ill health do not necessarily translate into days of lost work? How was it decided how many days of a ill patient will be considered as days lost?

Response 12: On page 10 this was mentioned “In estimating productivity losses, we followed recommended practice to use the actual income losses rather than the potential losses. The rationale is that, in agricultural societies or for people engaged in informal labour there are seasons in which work intensity is high and others in which work intensity is low. In addition, household use coping strategies that may mitigate these potential losses., Thus, days of ill-health do not necessarily translate neatly into days of lost work [29, 30]. We have further clarified this point in the text, as follows:

“Therefore, participants were asked how much money they and/or their caregiver had lost during the illness because of not participating in productive activities or income generation. The values reported by the respondent were considered to be indirect costs to the patient and caregiver.” PP10,line 195-198

13. Table 2. It is not clear what the household characteristics mean in terms of expenditure? How was expenditure differentiated by male or female?

Response 13: We have changed the title to “participant characteristics” and added sub-headings to make the categories clearer. The expenditure was based on a report obtained from the patient (either male or female) who had undergone surgical care.

14. All tables should report n for each specific item. For example, from table 3 it is not clear which expenditure item was applicable for how many patients. What happened if any item of expenditure was zero to an individual? How does this representative to actual scenario.

Response 14: We have now added median (IQR) and N to the table. All participants responded to ALL items. These expenditures are real zeros, not missing. PP15 line 276

15. As OOP expenditure for health tends to be highly skewed, Table 3 should also include the median expenditure.

Response 15: We have now added this.

16. The author should include equation for each of the estimation procedure to make it clear that how those estimates were derived.

Response 16: We have added this information as a supplementary file.

17. The author should also include clear explanation which of statistical test conducted for which of the variables. E.g., Table 6, not clear about the statistical inference.

Response 17: We have provided a footnote to the table indicating that Kruskal-Wallis test was used for comparing pre and post payment by surgical care sub-groups

18. P26, L426: The author mentioned that “Inequitable impacts of surgical care costs were evident in our study.” How the authors become sure that this was evident not due to the number of different types of surgical cases between the two regions?

Response 18: The findings underpinning this statement are presented in Table 7 (regression analysis).

19. The inclusion of subsidized cases and obstetric cases (the costs of which are also exempted) might have influenced the overall costs estimates per patient. In such cases, how useful it is to conclude the findings on the financial indicators without having a proper comparison of between subsidized cases and non-subsidized cases.

Response 19: We were interested to examine the economic impact of the existing healthcare and financing systems for surgical problems. As noted in response 10, even though obstetric cases were subsidised, there were hidden costs as well as opportunity costs that were important to examine in relation to catastrophic out of pocket healthcare expenditure.

20. Minor comments: the use of abbreviations is inconsistent throughout the manuscript.

Response 20: Thank you for your careful attention and for raising this point. We have now made it consistent throughout the text.

---

## [Decision Letter · Decision Letter 1]

15 Jul 2025

Dear Dr. Hanlon,

Thank you for submitting your manuscript to PLOS ONE. After careful consideration, we feel that it has merit but does not fully meet PLOS ONE’s publication criteria as it currently stands. Therefore, we invite you to submit a revised version of the manuscript that addresses the points raised during the review process.

We look forward to receiving your revised manuscript.

Kind regards,

Siddhesh Zadey

Academic Editor

PLOS ONE

Journal Requirements:

Reviewers' comments:

Reviewer's Responses to Questions

**Comments to the Author**

Reviewer #1: All comments have been addressed

Reviewer #2: All comments have been addressed

2. Is the manuscript technically sound, and do the data support the conclusions?

Reviewer #1: Partly

Reviewer #2: Yes

3. Has the statistical analysis been performed appropriately and rigorously?

Reviewer #1: I Don't Know

Reviewer #2: Yes

4. Have the authors made all data underlying the findings in their manuscript fully available?

Reviewer #1: Yes

Reviewer #2: Yes

5. Is the manuscript presented in an intelligible fashion and written in standard English?

Reviewer #1: Yes

Reviewer #2: Yes

Reviewer #1: I would like to thank the authors for addressing the comments. I have a few observations which might help to further clarify the manuscript.

1. The authors have mentioned that “Households were forced to use hardship coping mechanisms”, I was looking for if there is anything discussed in the manuscript on what is hardship coping mechanisms but did not find anything. This should be clarified.

2. I would suggest rather than mentioning just “Probably the frequent stock out of medicines and medical supplies may force patients to buy these items from private retail pharmacies”, the authors can check which OOP expenditure item was the highest for obstetric care and support the findings by reference previous literature.

3. The authors have clearly responded to the comment 10, however, the same explanation need to be included in the manuscript as potential reasons for OOP payments despite the care are free. I did not see similar explanation.

4. The author could also mention at somewhere how the obstetric care is free, is it the government policy or under insurance/protection scheme? If it is any insurance scheme who pays the premium?

5. The author could provide a supplementary table on the type of surgical patients they interviewed beside categorizing them by obstetric and non-obstetric patients. This will be useful to get information on the costs of different types of surgery and understand.

6. A few major costs items can be also provided in US$ for international comparison.

Reviewer #2: Thank you for the opportunity to review this manuscript. Comments raised by previous reviewers have been addressed well by the authors. They have clarified certain queries as well as acknowledged limitations to the study that were pointed out. The study objectives, methods, results, and conclusions are well aligned with each other. The methodology used is appropriate based on feasibility. All the financial outcomes, both direct and indirect, have been rigorously documented and analysed.

Apologies for the delay in my response.

**Do you want your identity to be public for this peer review?** For information about this choice, including consent withdrawal, please see our Privacy Policy

Reviewer #1: No

Reviewer #2: No

---

## [Author Response · Author response to Decision Letter 2]

4 Nov 2025

1st September 2025

Dr Emily Chenette

Editor-in-chief, PLoS One Journal

Dear Dr Chenette

Re: Revision of submitted manuscript

PONE-D-23-35112R1

Household costs, catastrophic out-of-pocket payments and impoverishment related to accessing surgical care in rural Ethiopia

We are grateful for the further comments from reviewers. Please see below for full responses to each reviewer point. Please let us know if you require any further clarifications.

Yours sincerely

Professor Charlotte Hanlon

REVIEWER COMMENT OUR RESPONSE

# Reviewer #1

1 The authors have mentioned that “Households were forced to use hardship coping mechanisms”, I was looking for if there is anything discussed in the manuscript on what is hardship coping mechanisms but did not find anything. This should be clarified

RESPONSE: Thank you very much for the valuable and constructive comments. Borrowing and selling assets are considered as hardship coping mechanisms. We clarified this in the following statement:

“Moreover, the other coping strategies used were borrowing and selling assets (i.e., hardship coping mechanisms), which are deeply regressive and could induce long-term household economic impacts and impoverishment” (pp27, lines 474-475)

2 I would suggest rather than mentioning just “Probably the frequent stock out of medicines and medical supplies may force patients to buy these items from private retail pharmacies”, the authors can check which OOP expenditure item was the highest for obstetric care and support the findings by reference previous literature.

RESPONSE: Table 4 provides the direct medical cost. For obstetric care, medicine, supplies and services accounted for 38.8% of this cost supporting our argument. We explained now this as ”Probably the frequent stock out of medicines and medical supplies in government facilities may force patients to buy these items from private retail pharmacies. This has been also reported in other studies [40.41]”

We have now included these references. PP 24 lines 395-396.

3 The authors have clearly responded to the comment 10, however, the same explanation need to be included in the manuscript as potential reasons for OOP payments despite the care are free. I did not see similar explanation.

RESPONSE: Responded above under No 2.

4 The author could also mention at somewhere how the obstetric care is free, is it the government policy or under insurance/protection scheme? If it is any insurance scheme who pays the premium?

RESPONSE: It is a government policy and this was stated on PP 5 lines 92-98

“In Ethiopia pre-payment and other financial risk protection mechanisms are limited. In recognition of this, in recent years the country has introduced health care financing reforms to increase affordable access to health services and achieve universal health coverage (UHC) [8]. Efforts to enhance financial risk protection for people accessing essential health services in Ethiopia include provision of high-impact interventions free of charge through an exemption program”.

Moreover, this was also mentioned on PP24 lines 377-379

In Ethiopia, surgical services are based on payments at the point of care, with the exception of obstetric services which are meant to be provided free of charge through the exemption programme.’

5 The author could provide a supplementary table on the type of surgical patients they interviewed beside categorizing them by obstetric and non-obstetric patients. This will be useful to get information on the costs of different types of surgery and understand.

RESPONSE: We have added a supplementary table as suggested and added the following text on PP13 lines 253-255.

“The frequencies of differing types of surgical operation are presented in Supplementary File 5: 66% were obstetric surgical procedures.”

6 A few major costs items can be also provided in US$ for international comparison.

RESPONSE: This was addressed in the foot note of the tables that provide costs. Showing 1 USD = 29.07 Birr (2019, purchasing power parity, 1 international dollars=10.74 Birr)

Reviewer #2:

1 Thank you for the opportunity to review this manuscript. Comments raised by previous reviewers have been addressed well by the authors. They have clarified certain queries as well as acknowledged limitations to the study that were pointed out. The study objectives, methods, results, and conclusions are well aligned with each other. The methodology used is appropriate based on feasibility. All the financial outcomes, both direct and indirect, have been rigorously documented and analysed.

RESPONSE: Thank you very much

---

## [Decision Letter · Decision Letter 2]

5 Jan 2026

Household costs, catastrophic out-of-pocket payments and impoverishment related to accessing surgical care in rural Ethiopia

PONE-D-23-35112R2

Dear Dr. Hanlon,

We’re pleased to inform you that your manuscript has been judged scientifically suitable for publication and will be formally accepted for publication once it meets all outstanding technical requirements.

Kind regards,

Siddhesh Zadey

Academic Editor

PLOS One

Additional Editor Comments (optional):

Reviewers' comments:

Reviewer's Responses to Questions

**Comments to the Author**

Reviewer #3: All comments have been addressed

2. Is the manuscript technically sound, and do the data support the conclusions?

Reviewer #3: Yes

3. Has the statistical analysis been performed appropriately and rigorously?

Reviewer #3: Yes

4. Have the authors made all data underlying the findings in their manuscript fully available?

Reviewer #3: (No Response)

5. Is the manuscript presented in an intelligible fashion and written in standard English?

Reviewer #3: Yes

Reviewer #3: The authors have adequately responded to the comments of the previous reviewers. I have no further comments.

**Do you want your identity to be public for this peer review?** For information about this choice, including consent withdrawal, please see our Privacy Policy

Reviewer #3: No

---

## [Editor Report · Acceptance letter]

PONE-D-23-35112R2

PLOS One

Dear Dr. Hanlon,

I'm pleased to inform you that your manuscript has been deemed suitable for publication in PLOS One. Congratulations! Your manuscript is now being handed over to our production team.

Kind regards,

on behalf of

Mr. Siddhesh Zadey

Academic Editor

PLOS One